# Identifying Biomarkers to Pair with Targeting Treatments within Triple Negative Breast Cancer for Improved Patient Stratification

**DOI:** 10.3390/cancers11121864

**Published:** 2019-11-26

**Authors:** Holly Tovey, Maggie Chon U. Cheang

**Affiliations:** Clinical Trials and Statistics Unit, The Institute of Cancer Research, London SM2 5NG, UK

**Keywords:** triple negative breast cancer, targeted therapy, molecular biomarkers

## Abstract

The concept of precision medicine has been around for many years and recent advances in high-throughput sequencing techniques are enabling this to become reality. Within the field of breast cancer, a number of signatures have been developed to molecularly sub-classify tumours. Notable examples recently approved by National Institute for Health and Care Excellence in the UK to guide treatment decisions for oestrogen receptors (ER)+ human epidermal growth factor receptor 2 (HER2)- patients include Prosigna^®^ test, EndoPredict^®^, and Oncotype DX^®^. However, a population of still unmet need are those with triple negative breast cancer (TNBC). Accounting for 15–20% of patients, this population has comparatively poor prognosis and as yet no targeted treatment options. Studies have shown that some patients with TNBC respond favourably to DNA damaging drugs (carboplatin) or agents which inhibit DNA damage response (poly ADP ribose polymerase (PARP) inhibitors). Known to be a heterogeneous population, there is a need to identify further TNBC patients who may benefit from these treatments. A number of signatures have been identified based on association with treatment response or specific genetic features/pathways however many of these were not restricted to TNBC patients and as of yet are not common practice in the clinic.

## 1. Introduction

Breast cancer is the most common malignancy diagnosed in the UK, with over 55,000 new cases diagnosed each year [1]. Traditionally, tumours are classified according to the presence of oestrogen receptors (ER), progesterone receptors (PgR) (considered together as hormone receptor status), and human epidermal growth factor receptor 2 (HER2). Treatment beyond surgery, chemotherapy, and radiotherapy is directed according to ER, PgR, and HER2 status, with endocrine therapy or trastuzumab available for patients with hormone receptor positive and HER2 positive tumours, respectively.

Accounting for approximately 10–20% of breast cancer diagnoses, triple negative breast cancers (TNBC) are characterised by ER, PgR, and HER2 negativity. These sub-classifications of breast cancer however mask further heterogeneity and classification beyond these well-established biomarkers can provide further information regarding prognosis for patients. A number of prognostic algorithms are available to predict patients’ risk of recurrence including Oncotype DX^®^, MammaPrint^®^, EndoPredict^®^, and Prosigna^®^. Many of these assays can also help to inform chemotherapy decisions for patients but other than MammaPrint^®^ are exclusively aimed at hormone receptor positive patients, with the picture for TNBC being less clear. A number of molecular subtypes within TNBC have been identified but as of yet there is no consensus on how these should be used to inform treatment choices for patients. Given the worse prognosis for these patients, there is an outstanding need to identify targeted treatment options to improve the likelihood of therapeutic success in TNBC. 

In this review we aim to summarise the current knowledge about promising targeted therapy for TNBC and associated molecular signatures for treatment response.

## 2. Molecular Heterogeneity within Triple Negative Breast Cancer

### 2.1. Intrinsic Subtypes

A number of attempts have been made to sub-classify breast cancer tumours to further explain the inherent heterogeneity (Table 1). One of the most renowned is the intrinsic subtypes first discussed in 2000 by Perou et al. [2]. Using hierarchical clustering of gene expression data from DNA microarray, Perou et al. identified a set of 496 genes, referred to as the “intrinsic gene subset”, which showed greater between than within sample variation. Using expression patterns of the intrinsic gene subset, it was shown that tumours could be classified into one of five intrinsic sub groups; Basal-like, HER2-enriched, Luminal A, Luminal B, and Normal-like [3]. In 2009, Parker et al. refined the intrinsic gene subset to an optimal list of 50 genes. A final classification algorithm based on these 50 genes, referred to as the PAM50 classifier, was established using nearest shrunken centroid methodology [4]. New samples are classified into an intrinsic subgroup based on the nearest centroid method. 

The intrinsic subtypes were observed to be highly associated with ER and HER2 status with the majority of triple negative tumours being classed as Basal-like [8,9]. Despite these associations, the intrinsic subtypes have been shown to be independent predictors of relapse free survival and neoadjuvant chemotherapy response in untreated and treated patients, respectively [4]. Given the majority of TNBC patients are classified as Basal-like, tremendous efforts have been made to molecularly dissect further the TNBC/non Basal-like tumours as well as to identify drug targets for Basal-like tumours.

More recently, an additional intrinsic subtype termed Claudin-low was discovered, characterised by high expression of epithelial-to-mesenchymal transition markers and low expression of claudins 3, 4, and 7 [5,10]. Gene expression profiles of Claudin-low tumours is similar to that of Basal-like tumours, with a key difference being lower expression of genes associated with proliferation [5]. Similar to the Basal-like subtype, Claudin-low tumours are most prevalently observed in TNBC but have slightly improved prognosis, although this does not reach statistical significance. Compared to the other intrinsic subtypes, response rates to anthracyclines/taxanes in Claudin-low tumours is lower than that of Basal-like tumours but still higher than Luminal A and Luminal B [5].

In 2011, Lehmann et al. used cluster analysis of gene expression profiles to identify six genetic subtypes within triple negative breast cancer; Basal-like one and two, Immunomodulatory, Mesenchymal, Mesenchymal Stem-like, and Luminal Androgen Receptor subtypes [11]. Similar to the intrinsic breast cancer subtypes, relapse free survival was significantly different between TNBC subtypes (*p* = 0.008) however distant metastasis free survival was not (*p* = 0.218) suggesting the relapse free survival difference is driven by a difference in local recurrence rates. Using TNBC cell lines, Lehmann et al. showed differential response rates between cell lines to different treatments. However, results were not always consistent for cell-lines representing a single subtype. For example, the *BRCA1* mutant cell line demonstrated a sensitivity to poly ADP ribose polymerase (PARP) inhibitors which was not found for all other cell-lines representing the Basal-like subtypes. They did however identify a difference in response rates to neoadjuvant taxanes in a meta-analysis of two studies, with preferential response rates in the Basal-1 and Basal-2 subtypes. In 2013, Masuda et al. also showed an association between pathological complete response rates and the Lehmann subtypes for 130 patients treated neo-adjuvantly with taxanes and/or anthracyclines [12]. Confirming the results shown by Lehmann’s group, the best response rates were seen in patients classified as Basal-1 [12]. These results highlight the potential to target neoadjuvant treatment with taxanes to those triple negative tumours classed as Basal-like. 

Lehmann et al. further refined the six subtypes to four, dropping the Immunomodulatory and Mesenchymal Stem-like subtypes after identifying that these subtypes had a large number of infiltrating lymphocytes or mesenchymal cells [6]. Using the refined subtypes, initially no significant differences in complete response rates to neoadjuvant chemotherapy were seen (regimens contained a taxane and/or anthracycline, results were consistent across regimens). A combined analysis of four datasets however showed that Basal-like one tumours had a significantly higher response rate compared to the other subtypes. Similar results were also found in a recent study by Echavarria et al. [13] in which RNA sequencing data from FFPE samples was available for 94 patients treated with neoadjuvant carboplatin and docetaxel. Pathological complete response rates were significantly associated with the refined Lehmann subtypes (*p* = 0.027) with the highest rate seen in Basal-1 patients with 65.6%, followed by 47.4% in Basal-2, 34.8% in Mesenchymal, and 21.4% in Luminal androgen receptor (AR) [13].

An eighty gene signature was published by Burstein et al. in 2015, classifying TNBC patients into one of four subtypes; Luminal-AR (LAR), Mesenchymal (MES), Basal-like Immune-Suppressed (BLIS), and Basal-like Immune-Activated (BLIA) referred to as the Baylor subtypes [7]. The subtypes showed significantly different disease free and disease specific survival with the worst and best prognoses observed for patients classified as Basal-like Immune Suppressed and Basal-like Immune Active, respectively. Substantial overlap with the intrinsic subtypes was observed with the BLIS and BLIA subgroups containing only Basal-like tumours whereas the LAR subgroup was a mix of Luminal A, Luminal B, and HER2-enriched. MES encompassed the remaining Basal-like tumours and included the Normal-like samples. Some concordance with the original Lehmann TNBC six subtypes was also observed, with good overlap of the LAR subtypes according to both classifications as well as the mesenchymal groups. Basal-like 1 and Basal-like 2 were both split between BLIA and BLIS indicating that the signatures are picking out different features within Basal-like tumours. 

A number of studies have been carried out to provide insight regarding racial disparity between subtypes. The Carolina Breast Cancer Study Phase III is a population-based study, within which the PAM50 algorithm was successfully applied to 980 white or African American breast cancer patients. Results showed that Basal-like tumours were more prevalent in African American women compared to white women [14], this held true across age groups (<50 versus ≥50). On the other hand, in the same study, Luminal A tumours were observed less frequently in African American women [14]. Jiang et al. looked at TNBC subtypes within a cohort of 360 Chinese women; compared to African American and Caucasian TNBC subsets from TCGA, the Chinese cohort had a significantly higher rate of Luminal AR tumours (*p* < 0.05) [15]. 

The disparities between these different breast cancer subtypes despite the generally good overlap serves to highlight the complexities of the heterogeneity within TNBC. Although all three subtypes provide prognostic information for patients, further work is required in order to be able to personalise therapy for TNBC patients.

### 2.2. Androgen Receptor Expression

Androgen receptor has been shown to be expressed in 12–55% of patients with triple negative breast cancer, although rates vary by study [16]. Prognosis of AR positive tumours within TNBC appears conflicting; studies have shown lower chemotherapy response rates in AR expressing tumours, likely due to the lower Ki67 rate in these tumours [17]. On the other hand, AR expression has also been associated with overall improved prognosis, as summarised by Gerratana et al. [17], although chemotherapy use in the studies is not reported. 

Although previously only considered relevant for Luminal Androgen Receptor subtypes which are largely characterised by AR expression, studies have shown that AR is also expressed in non-LAR subtypes [18]. Studies in breast cancer cell lines showed reduced proliferation and increased apoptosis in non-LAR lines when treated with the androgen antagonist enzalutamide, even when AR expression was low [16]. A clinical study of enzalutamide in patients with metastatic/locally advanced triple negative, AR positive (AR staining > 0%) breast cancer has also reported promising results. A clinical benefit rate of 33% was observed at 16 weeks in the evaluable population [19], therefore meeting the criteria for further study; other trials are ongoing.

TNBC tumours expressing AR have also been shown to be highly enriched for *PIK3CA* kinase mutations both in cell lines [11] and patient samples [20]. Following on from this finding, Lehmann et al. went on to show that PI3K inhibitors combined with AR targeting had an additive effect when applied in AR positive TNBC cell lines [20]. These results seem promising however pairing this treatment approach with AR status is yet to be confirmed by testing within a clinical trial.

### 2.3. Tumour Infiltrating Lymphocytes

A number of studies have examined the prognostic value of tumour infiltrating lymphocytes (TILs) in triple negative breast cancer. Across these studies, stromal TILs have been shown to be associated with outcomes in patients treated with adjuvant or neo-adjuvant chemotherapy [21,22,23,24,25]. These studies consistently showed that higher rates of stromal TILs were independently predictive of improved pathological complete response, disease free survival, and overall survival regardless of whether they are considered as continuous or categorised variables. Many of these studies did not specifically evaluate the effect in different chemotherapy regimens, however, Loi et al. showed that there was no significant interaction between stromal TILs and inclusion of taxanes (patients received either anthracyclines or anthracyclines plus a taxane) [22]. This suggests that stromal TILs may be predictive of general chemo-sensitivity in triple negative breast cancer. 

More recently, TILs have also been looked at in early stage TNBC patients who did not receive systemic therapy. A pooled analysis of four cohorts showed that the level of stromal TILs at diagnosis was prognostic in these patients when looking at invasive disease free survival, distant disease free survival, and overall survival [26] with better outcomes observed in those with higher levels of TILs. The study found that stromal TILs were associated with higher grade but not with other clinicopathological factors, therefore the prognostic effects were shown to be independent of other prognostic factors. Combined with the evidence from the earlier studies, stromal TILs look to be an ideal marker for identifying patients with good prognosis regardless of whether or not systemic therapy is used. Therefore, stromal TILs levels may identify a subset of patients in whom chemotherapy could be avoided without compromising outcomes.

## 3. Promising Targeted Therapy for TNBC

### 3.1. PARP Inhibitors

PARP inhibitors have been studied as an approach to cancer treatment for several years. As summarised by Plummer, the first PARP inhibitor was given as a chemo-potentiator in combination with chemotherapeutic agents in 2003 [27]. Since then, increased understanding of the mechanisms of action of PARP inhibitors and the different forms of DNA repair, has led to the approach of using them as a single agent in patients with deficient homologous recombination repair pathways. 

The rationale behind their use is the concept of synthetic lethality. PARP1 and PARP2 enzymes are involved in the DNA repair of single strand breaks. By impairing PARP1 and 2 via use of an inhibitor, the accumulation of single strand breaks can lead to double strand breaks. In the absence of functioning homologous recombination repair, such as in the presence of a *BRCA1/2* mutation, these double strand breaks cannot be fixed efficiently which results in cell death [28].

Since their first use in the early 2000s, PARP inhibitors have more recently been shown to be an effective maintenance therapy in women with newly diagnosed advanced ovarian cancer with a germline or somatic *BRCA1/2* mutation [29]. They have also been shown to be effective in women with HER2 negative advanced/metastatic breast cancer with an inherited *BRCA1/2* mutation [30,31] and have recently been approved by the US Food and Drug Administration (FDA) for use in this setting. Recent interim results from the PROfound study suggest these effects also hold true in prostate cancer patients with alterations in a number of homologous recombination repair genes beyond *BRCA1/2* [32]. This was a randomised phase III trial comparing the PARP inhibitor olaparib with physician’s choice of enzalutamide or abiraterone in men with metastatic castrate-resistant prostate cancer with an alteration in one of 15 genes involved in homologous recombination repair. An impressive hazard ratio of 0.49 (0.38 to 0.63) in favour of olaparib was seen for the primary endpoint of radiographic progression free survival [32].

### 3.2. Platinum Agents

Platinum agents such as carboplatin and cisplatin are used in cancer treatment due to their ability to cause DNA double stranded breaks through the formation of DNA inter-strand cross-links [33,34]. Several phase II and III studies have shown that the addition of platinum agents in the neoadjuvant setting can improve response rates in women with triple negative breast cancer [35,36,37,38]. The BrighTNess and GeparSixto studies went on to look at response rates according to germline BRCA mutation status and found no significant interactions between BRCA mutation status and treatment group [36,38]. Further to this, although no significant interaction was detected, a difference in response rates was observed in GeparSixto but this was in fact driven by improved response rates in the BRCA wildtype patients, with patients with a BRCA mutation achieving good response rates regardless of the treatment group assigned. In the advanced setting however, the TNT trial showed the opposite, with no benefit of carboplatin over docetaxel in the overall triple negative breast cancer population but a significantly improved response rate for carboplatin compared to docetaxel when analysis was restricted to those with a germline *BRCA1/2* mutation [39]. These contradictory results suggest that further exploration of the biology driving tumour response is required in order to identify the group of patients most likely to derive benefit from platinum-based chemotherapy.

PARP inhibitors and platinum agents have to date largely been focussed on patients with a *BRCA1/2* mutation. It is however hypothesised that a larger group of patients without *BRCA1/2* mutations but with other homologous recombination repair deficiencies could also benefit from these treatment approaches. Several groups are working on molecular biomarkers to identify these patients as outlined later in this review. 

### 3.3. CDK4/6 Inhibitors

Cyclin-dependent kinase (CDK) 4/6 inhibitors work by interrupting the cell-cycle to reduce proliferation of cancer cells. To date, three CDK4/6 inhibitors (palbociclib, ribociclib, and abemaciclib) have been approved by the FDA for use in patients with advanced/metastatic oestrogen positive, HER2 negative breast cancer following a number of successful trials in this disease setting [40]. Previously, triple negative breast cancers were not thought to be a good candidate for treatment with CDK4/6 inhibitors due to approximately 20% of these tumours lacking functional Retinoblastoma-like protein (Rb) [41]. Pre-clinical data however has indicated the potential for sensitive subtypes of TNBC; in particular, a study by Asghar et al. showed that the LAR subtype of TNBC was CDK4/6 inhibitor sensitive in vitro and in vivo [42]. Other TNBC tumours with high RB expression, androgen receptor positivity, or associated clinical characteristics are also considered potential candidates [43] and some pre-clinical research suggests a benefit of combination treatment including CDK4/6 inhibition [41]. A number of phase I or II studies of CDK4/6 inhibitors are ongoing within subsets of TNBC patients and results are awaited. 

### 3.4. Immunotherapy

This year, the FDA gave approval for the combination of atezolizumab (a PD-L1 targeting immunotherapy drug) with chemotherapy in triple negative breast cancer. The approval came following the phase III IMpassion130 trial which showed an improvement in progression free survival following the addition of atezolizumab to neoadjuvant nab-paclitaxel in untreated metastatic TNBC, with a hazard ratio of 0.80 (95% confidence interval: 0.69 to 0.92) [44]. When restricted to the subgroup of patients with PD-L1 positivity, the benefit of adding atezolizumab was observed to be even more pronounced with a hazard ratio 0.62 (95% confidence interval: 0.49 to 0.78). Interim analysis of overall survival did not show a statistically significant difference between treatment groups overall, but Kaplan Meier analysis suggested a longer median overall survival in those with PD-L1 positive tumours.

Interim results of the Keynote173 trial were also presented last year. These showed that high stromal TILs and PD-L1 were associated with improved pathological complete response and objective response rates in primary TNBC which had been treated with the immunotherapy pembrolizumab and neoadjuvant chemotherapy [45]. No comparison was made to a regimen excluding the immunotherapy, but combined with the results from IMpassion130 suggest that immunotherapies in TNBC could be effective in patients with PD-L1 positive tumours. The benefit of immunotherapies in patients without this marker however is more uncertain at present. 

High mutational burden has also been suggested as a potential indicator of immunotherapy sensitivity. Recent results from one cohort of the TAPUR study showed 37% disease control rate in patients with metastatic breast cancer with high tumour mutational burden treated with pembrolizumab [46]. Further evidence however is required to support the ability of this potential biomarker to direct treatment.

A number of other immunotherapy trials in TNBC are ongoing which will provide further insight, however many of these are in unselected patients (Table 2) and there is still a need for identification of robust biomarkers to predict benefit of immunotherapy. 

## 4. Identification of Molecular Signatures for Treatment Response

### 4.1. Homologous Recombination Deficiency (HRD)

Beyond *BRCA1/2* mutations, wider homologous recombination deficiency subgroups have been defined to identify a broader subgroup of patients who may benefit from specific treatment strategies. Loss of heterozygosity (LOH), telomeric allelic imbalance (TAI), and large-scale state transitions (LST) are independent measures of genomic instability each associated with BRCA mutational status [47,48,49]. Timms et al. showed that a combined score generated by taking the mean of the scores was better at identifying samples with homologous recombination deficiency than the individual scores [50], this is referred to as the homologous recombination deficiency (HRD) score. Within triple negative breast cancer patients, an association between HRD score or HR deficiency (defined as HRD score ≥ 42 or a *BRCA1/2* mutation) and pathological complete response to platinum agents has been observed [51]. However, a similar association between HR deficiency and response was also observed with anthracycline or taxane based neoadjuvant chemotherapy in a separate retrospective study [52]. Similar results were observed in the advanced setting [39] suggesting the HRD score is a prognostic marker within triple negative breast cancer patients and not predictive of response to a particular treatment.

More recently developed, HRDetect is a mutational signature model developed using lasso logistic regression to identify patients with homologous recombination deficiency [53]. Developed to identify patients with a BRCA deficiency the model has 98.7% sensitivity and was able to identify a number of patients with deficiencies which had not previously been picked up, classifying a larger cohort of patients who could benefit from BRCA/homologous recombination deficient targeted treatment strategies. 

Earlier this year, Staaf et al. published the results of applying the HRDetect signature to TNBC patients from the observational SCAN-B study in Sweden [54]. Of the 237 patients with evaluable samples, they found that 58.6% of TNBC patients had high HRDetect scores (defined as a score >0.7). HRDetect high tumours were enriched for Basal-like (PAM50 Basal-like and TNBCtype Basal-like 1) and Mesenchymal tumours. On the other hand, HRDetect low tumours were enriched for Luminal AR tumours and had more PAM50 non-Basal-like (mainly HER2 enriched and normal-like) tumours compared to the high tumours. Of the patients treated with standard of care adjuvant chemotherapy (regimens varied but fluorouracil, epirubicin, and cyclophosphamide ± taxane was common), those with high HRDetect were shown to have better outcomes as assessed by invasive disease-free survival. This led the authors to conclude HRDetect high tumours to be more chemo-sensitive than patients with low HRDetect scores [54].

Further to this, when calculated in samples from the personalised oncogenomics project, another observational study, the model was shown to be associated with improved outcomes in advanced breast cancer patients treated with platinum agents [55]. It should however be noted that the sample size in this study was small and further analysis in a larger prospective study is required to confirm these results. 

### 4.2. Mutational Signature

Substantial work has been carried out to characterise mutational signatures by whole genome and/or exome sequencing in cancer which reflect the different mutations which have occurred within a tumour. One particular mutational signature, referred to as signature 3, has been shown to be highly associated with the presence of *BRCA1* and *BRCA2* mutations [56,57] in breast and other tumour types. It was noted however that a number of cases without *BRCA1* or *BRCA2* mutations also exhibited high levels of signature 3. This led Polak et al. to explore the association of signature 3 with the wider homologous recombination repair pathway. They identified associations of the signature with epi-genetic silencing of *BRCA1* and mutation/methylation in other key genes from the homologous recombination pathway including *PALB2* and *RAD51C* [58].

Mutational signature 3 was used in the development of the HRDetect signature however to our knowledge has not been tested alone for prognostic of predictive ability to date. Consequently, there is little evidence regarding prognosis or predictive ability of this signature within TNBC.

### 4.3. Gene Expression Signatures

Several gene expression signatures related to DNA damage response have also been developed in an attempt to identify sub-populations of patients likely to derive benefit from therapeutic approaches. A number of methodologies have been employed based on association of gene expression data with either biological features related to DNA damage response or DNA damaging treatment sensitivity. 

Two promising signatures for treatment response that have come out of these approaches are the PARPi7 and BRCA1ness signatures. The first was published in 2012 by Daemen et al. [59] who identified a subset of genes for which transcriptional levels were associated with sensitivity to the PARP inhibitor olaparib across a number of breast cancer cell lines. From an initial list of 118 candidate genes taken from different DNA repair pathways, seven were taken forward into signature development and combined using the weighted voting algorithm to define the PARPi7 signature. When applied to unselected breast cancer patients who had not been treated with a PARP inhibitor, 8–21% of patients were predicted to be PARP inhibitor sensitive based on the signature, identifying a substantial proportion of patients who may benefit from this treatment approach. Based on biological features rather than treatment sensitivity, the BRCA1-like signature was developed to identify patients classed as BRCA1-like according to DNA copy number profiles [60]. Using diagonal linear discriminant analysis, 77 genes were identified which could classify samples between the BRCA1-like and non-BRCA1-like groups. In order to create a signature more utilisable in the clinic, the authors adapted the signature to be centroid based and a threshold was selected to give a high sensitivity of 96.7% and specificity of 73.1% in classifying patients.

These two signatures were subsequently applied to the 72 patients randomised to veliparib and carboplatin arm and the 44 HER2-negative controls within the I-SPY 2 breast cancer trial. The interaction between each biomarker and treatment group was statistically significant even after adjustment for hormone receptor status (*p*-values of 0.001 and 0.02 for PARPi7 and BRCA1ness respectively) [61]. This supports the notion of these two signatures being predictive of PARP inhibition sensitivity although the authors acknowledge that veliparib and carboplatin were given in combination so it cannot be determined whether the signatures are predicting sensitivity to the combination or one of the individual agents. Results also require validation in a larger dataset as sample size for these subgroup analyses was small. 

Given the known association between the Fanconi anaemia/BRCA pathway with DNA damage repair deficiencies, Mulligan et al. sought to develop a DNA damage repair deficiency (DDRD) assay based on the molecular characterisation of patients with Fanconi anaemia [62]. Using Affymetrix microarray they identified differentially expressed probe sets between patients with Fanconi anaemia and a set of patient controls. A number of breast cancer samples (*n* = 107) enriched for BRCA mutations were split into separate ER positive and negative datasets. Within each dataset, hierarchical clustering was applied and clusters representing the molecular processes associated with Fanconi anaemia were classed as DDRD positive, with remaining samples classified DDRD negative. The classified ER positive and ER negative datasets were then re-combined and a 44-gene expression signature was identified to accurately classify samples as DDRD positive or negative. The authors went on to show that the signature could predict response to fluorouracil, adriamycin, and cyclophosphamide (FAC) chemotherapy in the neo-adjuvant setting and fluorouracil, epirubicin, and cyclophosphamide (FEC) in the adjuvant. The signature could not however predict survival outcomes in an independent cohort of patients who did not receive cytotoxic chemotherapy. These results suggest the potential use of this signature to predict which patients may benefit from the addition of anthracycline based chemotherapy. 

The DDRD signature was subsequently successfully applied to 381 early TNBC patients treated with an adjuvant anthracycline containing regimen from the SWOG 9313 study. The signature was shown to be predictive of disease-free survival and overall survival, with high scores associated with improved outcomes independent of other prognostic factors [63]. The study also looked at stromal TILS density and found a positive correlation between this and the DDRD signature suggesting the potential for DDRD high tumours to be targeted with immune checkpoint inhibitors [63].

Adopting a slightly different approach based on chromosomal instability, Carter et al. correlated 10,151 genes with total functional aneuploidy across a number of pan-cancer datasets to develop the CIN70 signature [64]. A chromosomal instability score was calculated for each gene by summing the correlation rank of the gene across the datasets. The CIN70 signature is then composed of the top 70 genes with the highest CIN score; a simpler version was also created using the top 25 genes only (CIN25). The authors showed that the CIN signatures could be used to predict clinical outcome across a number of datasets including breast cancer patients and furthermore provided additional prognostic information above tumour grade alone. The CIN70 signature was also explored within the I-SPY2 trial where no significant interaction between the signature and treatment group was observed (*p* = 0.22 after adjustment for hormone receptor status) [61]. It therefore remains to be seen if high chromosomal instability, as determined by this signature, is targetable or simply prognostic across treatments as treatment specific data was not available in the original paper. 

### 4.4. Promise of Liquid Biopsies in Clinical Management

One emerging biomarker for prognosis is the evaluation of circulating tumour DNA (ctDNA). This is a non-invasive assessment method based on the detection of ctDNA which has been released from the tumour into the blood stream. Garcia-Murillas et al. looked at the use of ctDNA measured in blood at a single post-operative timepoint or from serial sampling to predict outcomes in early breast cancer unselected for hormone receptor status [65]. Presence of ctDNA within both the single time point and serial sampling could predict relapse across tumour types including within TNBC patients (*p* = 0.009 and 0.003) [65]. Sample size was small with just 11 and 13 TNBC patients with available samples for single time-point and serial sampling respectively, the results however are supported by other small studies restricted to TNBC patients with similar findings [66,67] suggesting the potential for ctDNA as a biomarker for relapse. What is currently less clear is whether ctDNA detection can be used to direct treatment. One trial trying to provide insights for this this is the cTRACK-TN trial (NCT03145961). Patients are followed up with serial ctDNA screening after completion of primary treatment, with randomization between pembrolizumab and observation in those with ctDNA detected prior to 12 months.

## 5. Conclusions

Over the last 20 years, increased availability and improvements in molecular profiling has uncovered the vast molecular heterogeneity present within TNBC. Molecular subtypes based on gene expression profiles have been identified and shown to confer vastly different risk profiles which may help inform decisions regarding chemotherapy use. 

Standard treatment approaches in patients with TNBC was previously limited to surgery with chemotherapy and/or radiotherapy, with a distinct lack of available targeted therapies. Treatment pathways however are now evolving with the recent approval of PARP inhibitors for patients with a *BRCA1/2* mutation and ongoing research into the use of CDK4/6 inhibitors and immunotherapies. 

A number of signatures predicting treatment response have also been developed for TNBC patients with some showing promising results in retrospective analyses. Many of these however still require validation within prospective trials in order to be brought forward into the clinic. With the advent of multi-omics technologies, more advanced computational approaches are being applied to integrate such high-dimensional biological data with patient outcomes to derive robust genomic signatures to inform better clinical management and next generation clinical trial designs.

## Figures and Tables

**Table 1 cancers-11-01864-t001:** Summary of breast cancer sub-classifications within triple negative breast cancers (TNBC).

Subtype		Key Features	Frequency in Early TNBC [5,6,7]	Anticipated Chemotherapy-Sensitivity
Intrinsic subtypes	Basal-like	Gene expression similar to basal-epithelial cells. High expression of proliferation genes. High overlap with TNBC and enriched for BRCA mutations.	39–54%	High
	HER2-enriched	High expression of HER2-regulated genes. Good overlap with oestrogen receptors (ER)-, human epidermal growth factor receptor 2 (HER2+) tumours.	7–14%	Intermediate
	Luminal A	Gene expression similar to luminal-epithelial cells. High expression of ER-related genes.	4–5%	Low
	Luminal B	Gene expression similar to luminal-epithelial cells. Expression of ER-related genes low compared to Luminal A tumours.	4–7%	Low
	Claudin-low	High expression of epithelial-to-mesenchymal transition markers and low expression of claudins 3, 4, and 7. Lower proliferation compared to Basal-like.	25–39%	Intermediate
	Normal-like	Similar expression to normal breast tissue.	1%	Low
TNBC subtypes	Basal-like 1	High expression of genes related to cell cycle, DNA damage response, and proliferation.	32–36%	High
	Basal-like 2	Increased expression of growth factor signalling related genes.	18–24%	Intermediate
	Mesenchymal	Increased expression of genes related to cell motility, differentiation, and growth. Absence of immune cells.	24–25%	Intermediate
	Luminal androgen receptor (AR)	Enrichment of pathways which are hormonally driven but typically hormone receptor negative. High expression of AR-related genes.	14–22%	Low
Baylor	Luminal AR	High expression of oestrogen regulated genes but typically negative by ER staining.	15–33%	Low
	Mesenchymal	High expression of genes from the following pathways: Cell-cycle, mismatch repair, and DNA damage.	17–28%	Intermediate
	Basal-like Immune Suppressed	Low expression of immune-related pathway genes.	29–31%	High
	Basal-like Immune Activated	High expression of immune-related pathway genes.	25–30%	High

**Table 2 cancers-11-01864-t002:** Selection of ongoing trials of pembrolizumab or atezolizumab in TNBC (source: ClinicalTrials.gov).

Setting	ClinicalTrials.Gov Identifier	Study Name	Treatment	Planned/Final Sample Size	Status
Adjuvant	NCT03036488	KEYNOTE-522	Pembrolizumab + chemotherapy vs placebo + chemotherapy	1174	Open no longer recruiting
	NCT02954874	Pembrolizumab in Treating Patients with Triple Negative Breast Cancer	Pembrolizumab versus observation	1000	Recruiting
	NCT03498716	IMpassion030	Atezolizumab + chemotherapy versus chemotherapy	2300	Recruiting
Neoadjuvant	NCT02620280	NeoTRIPaPDL1	Atezolizumab + chemotherapy versus chemotherapy	278	Open no longer recruiting
	NCT03639948	NeoPACT	Pembrolizumab + chemotherapy	100	Recruiting
	NCT03281954	Clinical Trial of Neoadjuvant Chemotherapy with Atezolizumab or Placebo in Patients with Triple-Negative Breast Cancer Followed After Surgery by Atezolizumab or Placebo	Atezolizumab versus placebo	1520	Recruiting
	NCT02530489	Nab-Paclitaxel and Atezolizumab Before Surgery in Treating Patients with Triple Negative Breast Cancer	Atezolizumab + chemotherapy	37	Recruiting
Metastatic/locally advanced	NCT02819518	KEYNOTE-355	Pembrolizumab + chemotherapy vs placebo + chemotherapy	882	Open no longer recruiting
	NCT03121352	Carboplatin, Nab-Paclitaxel and Pembrolizumab for Metastatic Triple-Negative Breast Cancer	Pembrolizumab + chemotherapy	30	Open no longer recruiting
	NCT02555657	KEYNOTE-119	Pembrolizumab versus chemotherapy	622	Open no longer recruiting
	NCT02447003	KEYNOTE-086	Pembrolizumab	285	Open no longer recruiting
	NCT03125902	IMpassion131	Atezolizumab + chemotherapy versus placebo + chemotherapy	600	Recruiting
	NCT03371017	IMpassion132	Atezolizumab versus placebo	350	Recruiting
	NCT02734290	Standard of Care Chemotherapy Plus Pembrolizumab for Breast Cancer	Pembrolizumab + chemotherapy	88	Recruiting
	NCT03206203	Carboplatin with or without Atezolizumab in Treating Patients With Stage IV Triple Negative Breast Cancer	Atezolizumab + chemotherapy versus chemotherapy	185	Recruiting

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
