# Peer review of "Identifying Biomarkers to Pair with Targeting Treatments within Triple Negative Breast Cancer for Improved Patient Stratification"

_cancers, 2019, doi:10.3390/cancers11121864_

Round 1

Reviewer 1 Report

In the review, Tovey and colleagues highlighted some of the recent advances and outstanding challenges with targeted therapy for triple negative breast cancer (TNBC) and with the identification biomarkers/molecular signatures for predicting the response to therapy. Overall, this is an important area of research and there is an urgent need for development of novel and more effective and less toxic therapeutic interventions for patients diagnosed with TNBC.

My minor comments. This review would be strengthen, if the authors could also include the followings summary tables/figures:

A summary table/Figure of various therapeutic strategies for TNBC and their advantages and disadvantages A summary table of different approaches and their potentials for the identification of molecular signatures/biomarkers for predicting the response to therapy.   Table 2 should be formatted correctly (i.e. status column)

Author Response

My minor comments. This review would be strengthen, if the authors could also include the followings summary tables/figures:

Point 1: A summary table/Figure of various therapeutic strategies for TNBC and their advantages and disadvantages. A summary table of different approaches and their potentials for the identification of molecular signatures/biomarkers for predicting the response to therapy.  

Response 1: We thank the reviewer for their comment however we respectfully disagree. The topics are discussed in detail in the manuscript and we do not feel that repeating the information on a superficial level within a table would add significantly to the manuscript.

Point 2: Table 2 should be formatted correctly (i.e. status column) 

Response 2: We thank the reviewer and have corrected this.

Reviewer 2 Report

In this manuscript the authors provide a summary of the various transcriptional and prognostic signatures with respect to TNBC.  The manuscript is quite thorough and discusses promising targeted therapy emerging for the treatment of TNBC.

1.There is no mention of normal-like intrinsic subtype in table 1. Please add to table or just discuss in text.

2. The ranges of the intrinsic subtypes range from 79%-119% when added up. How were these determined (specific datasets, literature searches)?

3.Please add “absence of immune cells” to “Mesenchymal” subtype of “TNBC subtypes” in table per PMID:27310713.

4. Include a discussion of TNBC subtypes in primary Chinese population (PMID:30853353) in the “Intrinsic subtype” section.

5. Please include discussion of additional analysis of chemotherapy and TNBC subtypes (PMID:23948975 and PMID:29378733).

6. Please correct spelling of “Lehman” to “Lehmann” throughout the manuscript.

7. Include TBCRC043 NCT03206203 in metastatic immunotherapy section of table 2.

8. The author states, “A clinical study of enzalutamide in patients with triple negative, AR positive (AR staining >0%) breast cancer has also reported promising results [15] with other trials ongoing”. Please describe results of the AR clinical trial in more detail. What were the results or why are they promising.

9. Include some discussion on the enrichment of activating PIK3CA mutations in AR positive TNBC and the potential for targeting.

10. Please discuss PMID:31570822 in HRD section. Include HRDetect mutational signature with respect to other subtypes (Molecular apocrine, IC4 and LAR are enriched in HRD-low while IC1, basal like, BL1 and M enriched in HRDect high tumors.

11. Modify table 2 so that treatment and status do not overhang

12. Perhaps add a section discussing the potential of targeting NOTCH alterations in TNBC (PMID: 25104330, PMID:25564152 and PMID:22101766).

13. The discussion of the p53 mutation signature that was generated in ER+ cell lines at the end of the manuscript is somewhat outdated given that majority (80-90%) of TNBC have loss of p53 function. Especially since the signature has not been derived or tested in a TNBC setting. Consider removing or justifying the section as it does not end strong or fit with the theme of DNA damage response signatures in that section.

Author Response

In this manuscript the authors provide a summary of the various transcriptional and prognostic signatures with respect to TNBC.  The manuscript is quite thorough and discusses promising targeted therapy emerging for the treatment of TNBC.

Point 1. There is no mention of normal-like intrinsic subtype in table 1. Please add to table or just discuss in text.

Response 1. We thank the reviewer for this comment and have added the normal-like intrinsic subtype to table 1.

Point 2. The ranges of the intrinsic subtypes range from 79%-119% when added up. How were these determined (specific datasets, literature searches)?

Response 2.  The ranges of frequencies have come from the respective papers, to make this clearer we have added the references for these papers to the column header for frequency in table 1.

Point 3. Please add “absence of immune cells” to “Mesenchymal” subtype of “TNBC subtypes” in table per PMID:27310713.

Response 3. We thank the reviewer for their comment and have added this to table 1.

Point 4. Include a discussion of TNBC subtypes in primary Chinese population (PMID:30853353) in the “Intrinsic subtype” section.

Response 4. We thank the reviewer for their comment and have added a paragraph about racial disparities in subtypes to the end of the intrinsic subtypes section.

Point 5. Please include discussion of additional analysis of chemotherapy and TNBC subtypes (PMID:23948975 and PMID:29378733).

Response 5. We thank the reviewer for their comment and have accordingly added in some information about these studies to the relevant section on lines 87 & 99.

Point 6. Please correct spelling of “Lehman” to “Lehmann” throughout the manuscript.

Response 6. We thank the reviewer for pointing this out and have updated throughout the paper.

Point 7. Include TBCRC043 NCT03206203 in metastatic immunotherapy section of table.

Response 7. We thank the reviewer for their comment and have added this study to table 2.

Point 8. The author states, “A clinical study of enzalutamide in patients with triple negative, AR positive (AR staining >0%) breast cancer has also reported promising results [15] with other trials ongoing”. Please describe results of the AR clinical trial in more detail. What were the results or why are they promising.

Response 8. We thank the reviewer for their comment and have added further details of this study in the relevant section to line 147.

Point 9. Include some discussion on the enrichment of activating PIK3CA mutations in AR positive TNBC and the potential for targeting.

Response 9. We thank the reviewer and have added discussion around this to the AR expression section on line 150.

Point 10. Please discuss PMID:31570822 in HRD section. Include HRDetect mutational signature with respect to other subtypes (Molecular apocrine, IC4 and LAR are enriched in HRD-low while IC1, basal like, BL1 and M enriched in HRDect high tumors.

Response 10. We thank the reviewer and have added a brief summary of this study to the HRD section. On reflection however we have respectfully not included reference to the IC classifications as these are not covered in other sections of the paper.

Point 11. Modify table 2 so that treatment and status do not overhang

Response 11. We thank the reviewer and have corrected this.

Point 12. Perhaps add a section discussing the potential of targeting NOTCH alterations in TNBC (PMID: 25104330, PMID:25564152 and PMID:22101766).

Response 12. We thank the reviewer for their suggestion but have decided not to add this to the manuscript as we do not feel it will strengthen the paper.

Point 13. The discussion of the p53 mutation signature that was generated in ER+ cell lines at the end of the manuscript is somewhat outdated given that majority (80-90%) of TNBC have loss of p53 function. Especially since the signature has not been derived or tested in a TNBC setting. Consider removing or justifying the section as it does not end strong or fit with the theme of DNA damage response signatures in that section.

Response 13. Upon reflection we agree with the reviewer and have removed the section in question as it does not add to the manuscript.